# Effect of Time and Frequency of Magnetic Field Application on MRF Pressure Performance

**DOI:** 10.3390/mi13020222

**Published:** 2022-01-29

**Authors:** Purwadi Joko Widodo, Eko Prasetya Budiana, Ubaidillah Ubaidillah, Fitrian Imaduddin, Seung-Bok Choi

**Affiliations:** 1Department of Mechanical Engineering, Faculty of Engineering, Universitas Sebelas Maret, Surakarta 57126, Indonesia; purwadijokow@staff.uns.ac.id (P.J.W.); ekoprasetya@staff.uns.ac.id (E.P.B.); fitrian@ft.uns.ac.id (F.I.); 2Department of Mechanical Engineering, The State University of New York, Korea (SUNY Korea), Incheon 21985, Korea; 3Department of Mechanical Engineering, Industrial University of Ho Minh City, 12 Nguyen Van Bao Street, Vap District, Ho Chi Minh City 70000, Vietnam

**Keywords:** magnetorheological fluid (MRF), pressure, frequency, electric current

## Abstract

This research was conducted to determine the effect of the time and frequency of magnetic field application on MRF pressure performance. It was carried out by placing magnetorheological fluid (MRF) in a U-shaped, glass tube and then repeatedly applying a magnetic field to it for a certain time period with a particular frequency set by the generator frequency. The length of the application period of the magnetic field, the frequency of the application of the magnetic field, and the magnitude of changes in fluid pressure that occurred and changes in pressure in the MRF were recorded with a data logger for a specific time, which was 60 s. From the field tests that were carried out, it was found that during the application of a continuous magnetic field, there was pressure on the MRF until it reached the maximum pressure; then, there was a gradual decrease in pressure when the magnetic field was turned off, but the pressure was intense. It was shown that the pressure decreased rapidly as the magnetism disappeared, even causing the pressure to drop below the initial pressure, which, in turn, gradually rose again toward the equilibrium pressure. Meanwhile, during the repeated application of a magnetic field, it appeared that the MRF effectively produced pressure in response to the presence of a magnetic field up to a frequency of 5 Hz. The higher the applied magnetic field frequency, the smaller the pressure change that occurred. Starting at a frequency of 10 Hz, the application of a magnetic field produced more minor pressure changes, and the resulting pressure continued to decrease as the liquid level decreased toward the initial equilibrium position.

## 1. Introduction

Technology is developing quickly and continuously, which means the quality of human life is continually improving. The amount of attention paid to intelligent automation technology is relatively high, and the achievements that humans have made in this field with the support of current science and technology are very significant [1]. The fourth industrial revolution, known as Industry 4.0, has changed the future of the manufacturing industry. The manufacturing industry has widely adopted automation technology in place of conventional mechanical processes through the use of intelligent production automation processes [2].

The presence of intelligent materials influences intelligent automation technology. Intelligent materials are used to construct these intelligent structures, which can perform both sensing and actuation functions [3]. “Intelligent materials” is a general term used for a large group of different substances. A common feature of intelligent materials is that they have a large number of properties that can be significantly changed under controlled conditions. Initially, intelligent materials were defined as materials that can promptly respond to their environment. However, later on, the definition of an intelligent material was expanded to encompass a material that can receive, transmit, or process a stimulus and responds by producing a beneficial effect, including a signal indicating that the material responds to input [4]. Many electroactive functional materials have been used in small- and micro-scale transducers and precision mechatronic control systems over the years. However, it was only in the mid-1980s that scientists began to integrate electroactive materials with large-scale structures such as sensors and/or actuators, which meant that the concepts of intelligent materials, smart structures, and structural systems became more well known [5]. Some examples of intelligent materials are piezoelectric materials, shape memory alloys, magnetostrictive materials, electro-rheological materials, magnetorheological fluids, and thermoresponsive and electrochromic materials [6].

Magnetorheological fluid (MRF) is one of the many intelligent materials that work by responding to the influence of a magnetic field. This material can respond to the influence of a strong magnetic field by changing the position of its magnetic particles to form a chain structure in the direction of the magnetic field trajectory. Magnetorheological-material-based actuators have been an interesting research topic for more than half a decade. Many actuators have been developed based on magnetorheological fluids and magnetorheological elastomers such as dampers, brakes, haptic devices, couplings, and mounts [7]. Since Rabinow published the discovery of MRF in 1948 at the US National Bureau of Standards [8], various patents have been registered, and various research articles related to MRFs have continued to be published [9].

Various research articles and studies that take advantage of the potential of MRF continue to be carried out, for example, with regard to shock absorbers [7,10,11,12,13], brakes [14,15,16], valves [12,17,18,19], polishing [20,21,22], haptic devices [23,24,25], mounts [26,27,28], and sound propagation [29,30].

According to some research that has previously been carried out, MRF is an excellent candidate to replace conventional fluids in liquid-based systems. MRF-based systems can improve the performance and functionality of control systems in many applications, especially in terms of controllability, responsiveness, and wide application. A controllable, MRF-based system can provide more precise output control due to the variable viscosity of MRF and the fact that it switches between semisolid and fluid phases with the application of a magnetic field. In addition, the fast response reaction of an MRF-based system to an applied magnetic field is on the scale of milliseconds, making it a suitable candidate for use in real-time control applications. MRF-based control systems have been comprehensively used in many applications in various industries, including civil, aerospace, and automobiles, to enhance system performance and functionality to achieve the desired outputs [9].

Most research on MRF has looked at and developed the potential of MRF based on its ability to change its shear strength dynamically by exploiting the dynamics of changing electrically generated magnetic fields. These changes will then cause changes in the fluid shear strength. Changes in the shear strength will increase the shear resistance, and the power will increase its ability to resist external forces. This feature is utilized in most applications that apply MRF as a smart material. Another issue with less attention from this MRF potential is the ability of the carrier fluid to generate pressure as a result of the movement of magnetic particles in it. Therefore, the potential of the MRF to produce an action force could be challenging for further study. Regarding previous related work, investigations on the use of MRF as the main actuator are very minor. Some studies have elaborated on MRF for actuators, but they still needed a pumping system, as can be found in references by [31,32].

In previous studies, observations have been made with regard to changes in the magnitude of MRF pressure as a result of variations in the application of magnetic field strength to MRF [33]. From these observations, it can be seen that the phenomenon of a change in pressure strength associated with a change in the magnitude of magnetic field strength, which is represented in the form of a robust electric current applied to a magnetic coil, exists. In repeated cycles of magnetic field applications, an interesting phenomenon is seen: the emergence of changing pressures that occur in subsequent cycles. This is an interesting phenomenon that can be studied further. Therefore, in this study, further efforts were made to observe this phenomenon by providing a stronger current over a longer time period and by applying an electric current with a particular frequency to a magnetic coil to determine its effect on the MRF pressure performance. As in previous studies [33], this study was carried out using a U-type glass channel, which can easily be used to identify pressure changes by visually observing the difference in the liquid level. In addition, as a continuation of this research, a signal generator was used to adjust the frequency.

## 2. Materials and Methods

In this study, MRF-122EG was used as the test material. MRF was placed in a U-type conduit, where a coil was provided to generate a strong magnetic field. The coil was made of copper wire coil with a diameter of 0.5 mm, a total of 950 turns, and a coil height of 35 mm. The U-type channel made of Pyrex glass had a hole diameter of 5 mm, an outer diameter of 8 mm, a channel axis distance of 50 mm, a straight-line height of 100 mm, and supports made of PVC material, as shown in the illustration of Figure 1. MRF-122EG is a hydrocarbon-based magneto-rheological (MR) fluid formulated for general use in controlled energy dissipation applications such as shocks, dampers, and brakes. Table 1 shows the data properties of MRF that was used in this study.

This U-type tube was used because of its simplicity in measuring the pressure changes. Simply put, the difference in pressure present on the tube feet results in different fluid levels. Therefore, it indicates a pressure difference in the U-type channel. In the absence of a magnetic field, the two legs of the U-type show a balanced position, which indicates the balance of pressure in the two legs. A magnetic field is applied to the coil in the U-type channel by increasing the input electric current to indicate a change. In addition, the diameter of the channel bore has a significant effect on the MRF pressure strength. This is because the pressure resulting from the thrust generated by the movement of magnetic particles is divided by the cross-sectional area of the channel.

The coil was given a 12 V DC voltage with a robust electric current of 0.5 A, 1.0 A, and 1.5 A and varying application frequencies for 60 s. Conditions of static application, on–off application, and application with frequencies between 0.05 Hz, 0.1 Hz, 0.5 Hz, 1 Hz, and 10 Hz were all studied. Furthermore, a 12 V DC voltage with the 1.5 A current was used when studying the higher frequencies of 20 Hz, 40 H, 50 Hz, and 100 Hz. The current of 1.5 A was used for experiments that took place at a higher frequency, considering that a current of 1.5 A would have a more visible effect on pressure than a lower current strength; therefore, the pressure that did appear would be better represented.

A signal generator set the frequency. This signal generator was used to generate a square signal for 60 s, which was used to turn the mosfet switch on and off. In addition to sending a box signal, the signal generator also gave instructions to the data logger to start and end data recording in order for data recording to be carried out consistently. The mosfet switch was used to turn the current flow to the coil off and on. An ACS712 current sensor [35] was used to detect the current flow and send it to the data logger to be recorded by a computer.

In addition, the pressure generated by the MRF at the end of the U-shaped pipe was detected using the MPX5010DP pressure sensor [36] and sent to the data logger to be recorded by the computer. The data logger used CuteCom software and an Arduino that was configured to sample data readings from sensors. Figure 2 and Figure 3 show the block diagram and configuration of the test equipment used in the study.

Due to the fact that it is not easy to measure magnetic field strength in the research configuration, in this case, the magnetic field strength is represented in the form of an electric current. As has been mentioned in several articles [7,16,37,38], the magnitude of magnetic field strength in a magnetic coil can be predicted using FEMM software. Therefore, in this case, the strength of the magnetic field was directly proportional to the electric current flowing in the magnetic coil.

## 3. Result

Figure 4 provides an overview of the magnetic field conditions in the U-type channel in this study. The simulation results using FEMM software as shown in the figure was from applying a current of 1.5 A. Here, the simulation was carried out under the same condition as the experiment such as the use of copper wire AWG24 having of diameter 0.5 mm and a coil with 950 turns. The dimensions of the U tube in the simulation were also similar to those of the experiment.

Based on the data logger recordings, the observations that were obtained are presented in Table 2 and Table 3. In this case, two stages of observation were carried out, namely observations with frequencies below 10 Hz and follow-up observations for frequencies above 10 Hz. Observations under 10 Hz were carried out using three variations in current strength, namely 0.5 A, 1.0 A, and 1.5 A, with variations in application time, namely on–on status for approximately 60 s and on–off status for 30 s each, with frequencies of 0.05 Hz, 0.1 Hz, 0.5 Hz, 1 Hz, and 10 Hz, with each application time of 60 s. The results of the data logger recordings are presented in Table 2.

In the following observation stage, a 1.5 A current was used with higher frequencies, namely 20 Hz, 40 Hz, 50 Hz, and 100 Hz. The results are shown in Table 2. Although these observations were intentionally carried out using a 1.5 A current, it was expected that the pressure change phenomenon would be more visible with a larger current.

From the results of the first stage of observation, it can be seen that the pressure strength is related to the strength of the applied magnetic field, which in this case, is represented in the form of an electric current. In the long period of testing, as shown in Figure 5, it can be seen that when a magnetic field is applied, there is an increase in pressure, but the pressure decreases gradually. Meanwhile, when the magnetic field is removed, the pressure drops to a value even lower than the initial pressure, increasing slowly until the liquid level is at equilibrium. This condition is more clearly seen, for example, at a frequency of 1 Hz and with a 1.5 A current input. Furthermore, it is also seen at a frequency of 5 Hz with a strong current of 1.0 A. The pressure fluctuations increase with frequency, and it is because the movement of magnetic particles due to the application of a magnetic field that is getting faster cannot be responded to by the fluid to produce pressure. Therefore, based on the data, a pressure commensurate with the given current is produced. The maximum is obtained at a frequency of 1 Hz.

This condition was repeated at a frequency of 1 Hz magnetic field application, as shown in Table 2, but it was shown that the effectiveness of magnetic field application decreases its effect on pressure when the frequency reaches 5 Hz. Finally, at a frequency of 10 Hz, the effect of applying a magnetic field to the emergence of pressure is even more negligible.

In the advanced stage, namely observations regarding the application of a strong current of 1.5 A with a frequency above 10 Hz, the results obtained due to the application with such a high frequency show pressure at the beginning of the MRF, which is given as an electric current. Meanwhile, during the application of advanced electric current, the effect on pressure is miniscule. This is because the higher the frequency, the smaller the effect on the pressure changes that occur, and it was shown that under these conditions, the pressure continues to decrease until the current application is stopped, as shown in Table 3.

At a frequency of 20 Hz, it can be seen that the ability of the MRF to respond to a magnetic field to generate pressure becomes smaller. The pressure change generated by the MRF due to the application of a magnetic field becomes smaller. Additionally, the pressure continues to decrease until the end of the observation. The pressure changes that occur continuously become smaller at the frequencies higher than this.

## 4. Discussion

From the comparison of the application of current strength in Table 2, it is shown that the magnitude of the current applied to the MRF correlates with the resulting liquid pressure, as has been disclosed in previous studies [33]. In addition to the strong electric current, the amount of pressure is also influenced by the accumulation of magnetic particles on one side of the channel. This is indicated by the condition of the MRF face height, which is no longer parallel between the MRF face height on the left and right side of the channel. As the magnetic field is activated through the magnetic coil on the right side of the channel, the magnetic particles are attracted toward the right side of the channel. When the magnetic field is removed, the magnetic particles drop due to the gravitational force but do not spread evenly again as before. Thus, when the following magnetic field is applied, it causes magnetic particles to accumulate on one side of the channel used. The pressure that appears at the beginning where the magnetic particles are still evenly distributed differs from the pressure that occurs when the magnetic particles accumulate. Likewise, the occurrence of deposition when there is no solid magnetic force acting for a particular time can cause a loss of pressure in the fluid.

The phenomenon of separation between magnetic particles and the carrier fluid occurs when using MRF in squeeze mode [39,40], where, when a magnetic field is applied, the magnetic particles are held in position by the attraction of the field. However, the carrier liquid slowly falls back down due to the gravitational attraction, and this causes a gradual decrease in pressure, as shown in Figure 4, even though the current strength is maintained for a certain period because the carrier fluid falls back to the equilibrium position due to the gravitational force. As for when the magnetic force is removed, the liquid is also pushed by the magnetic particles, which causes the pressure to drop to values below the initial pressure. Then, it rises back up to the initial equilibrium position, shown in Figure 4, and this phenomenon is more clearly visible in repeated applications such as those at a frequency of 1 Hz with a current of 1.5 A or 5 Hz with a current of 1.0 A; in these applications, it appears that the condition of the curve decreases in the following cycles.

In providing a cyclical magnetic field, the above conditions were repeated by the given frequency of observations at frequencies below 5 Hz. At a frequency of 10 Hz, the effect of strong currents on the pressure that occurs was shown to be smaller. The faster the frequency used, the smaller the effect of a magnetic field on changes in fluid pressure, as shown at frequencies above 20 Hz, 40 Hz, 50 Hz, and 100 Hz.

Under these conditions, the MRF can respond well to the application of a magnetic field with a high-frequency MRF to produce a change in pressure, and the pressure even continues to decrease. This occurs because the liquid level continues to fall toward equilibrium. With a high frequency of magnetic field application, the change in the position of the magnetic particles is not too large. It results in an insignificant push against the liquid, so the change in pressure that occurs is not significant enough, while the liquid, as a result of gravity, continues to fall downwards toward the equilibrium position of the liquid surface.

With this experiment, the response of fluid motion can be seen as a result of the movement of magnetic particles. In applying a high-frequency magnetic field, the fluid cannot respond to changes in magnetic field strength appropriately. Instead, it can be seen from the ability of the fluid to produce pressure that decreases or even disappears. In some cases of applicable research, these results are quite influential. One example is the MR damper. The field-dependent damping force and response time to the input current can be easily expected by undertaking a simple test performed in this work. In addition, the pressure loss of the application system can be understood by observing the pressure change with respect to the frequency and current.

## 5. Conclusions

With the results of this study, it is known that with the stability of the MRF pressure performance with time and the frequency of application of the magnetic field, it appears that the pressure generated by the MRF is unstable, changes with time, and the MRF is unable to respond to generate pressure at the frequency of application of the magnetic field above 10 Hz.

The strength of the magnetic field, which is represented by the strength of the electric current, is correlated with the pressure of the resulting liquid. The pressure caused by the liquid is driven by the movement of magnetic particles that react in the presence of a magnetic field, and the stronger the magnetic field, the stronger the impulse, which results in the greater pressure that appears.

As a result of the separation of magnetic particles with the carrier, fluid causes the accumulation of magnetic particles on one side of the channel, affecting the pressure strength of the resulting liquid. The separation phenomenon is also seen in the application of strong currents over a long period; it is seen that the pressure that appears gradually decreases as a result of the separation of magnetic particles and liquid, where the liquid returns to the equilibrium position of the liquid surface.

During the application of a magnetic field with a certain frequency, these conditions were repeated, and the MRF was still shown to be effective in producing pressure up to a frequency of 5 Hz. At a frequency of 10 Hz, it was shown that the application of current was no longer effective in producing changes in fluid pressure.

At a higher frequency, the MRF’s ability to generate pressure is smaller; this can be seen in the 40 Hz frequency condition, where the pressure appeared at the beginning of the current supply, and then the pressure slowly continued to decrease until the magnetism was removed. Furthermore, further research needs to be performed to produce a more stable MRF pressure so that the MRF can be used as an active actuator. One of the things that makes the resulting pressure unstable is the separation between the particles and the carrier liquid. In addition, the influence of MRF densities is believed to have a different response instead of the long-term observation of characterization in this mode.

## Figures and Tables

**Figure 1 micromachines-13-00222-f001:**
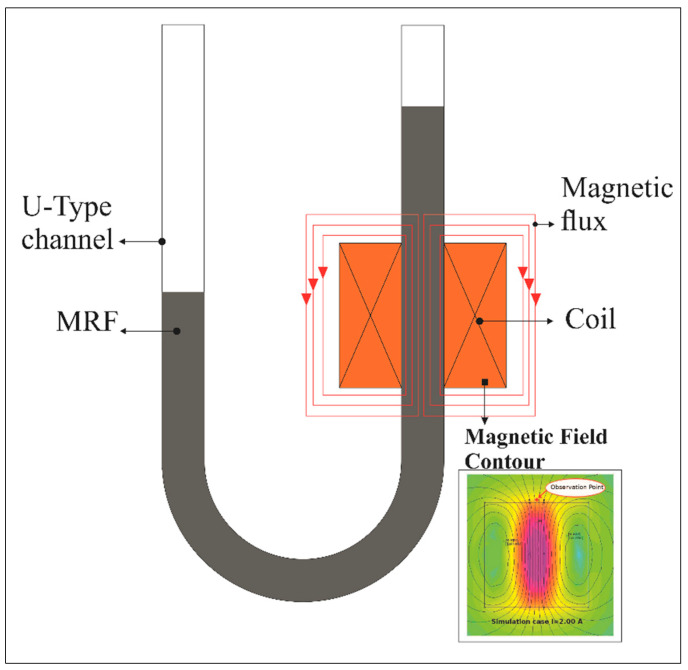
A schematic U-type channel and magnetic coil used for testing.

**Figure 2 micromachines-13-00222-f002:**
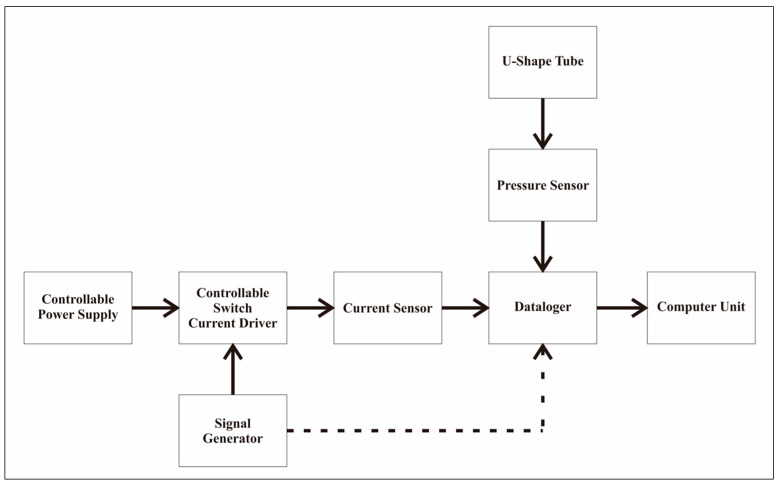
A schematic block-diagram for pressure measurement.

**Figure 3 micromachines-13-00222-f003:**
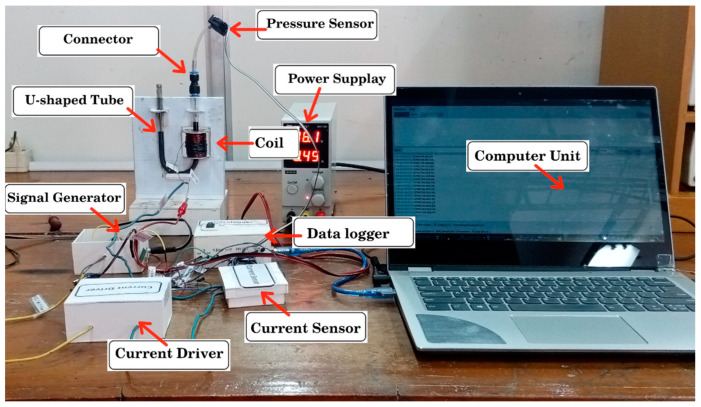
An experimental apparatus for the pressure measurement.

**Figure 4 micromachines-13-00222-f004:**
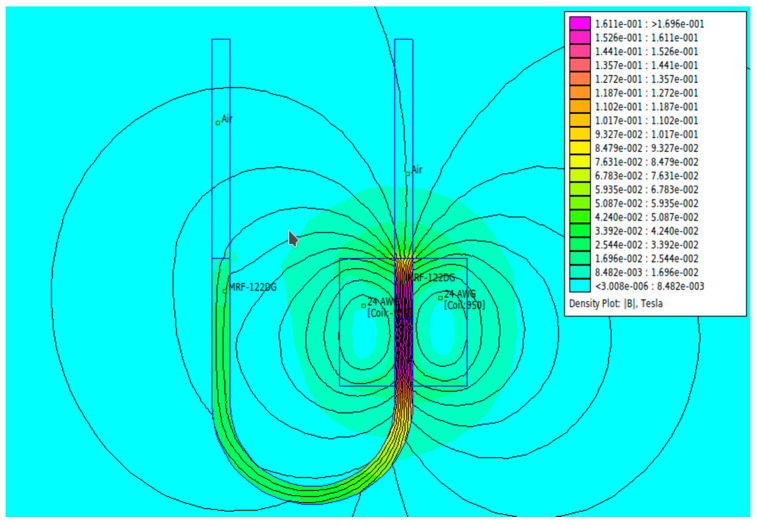
The FEMM simulation results: the magnetic field conditions in the U-type channel when a current of 1.5 A is applied.

**Figure 5 micromachines-13-00222-f005:**
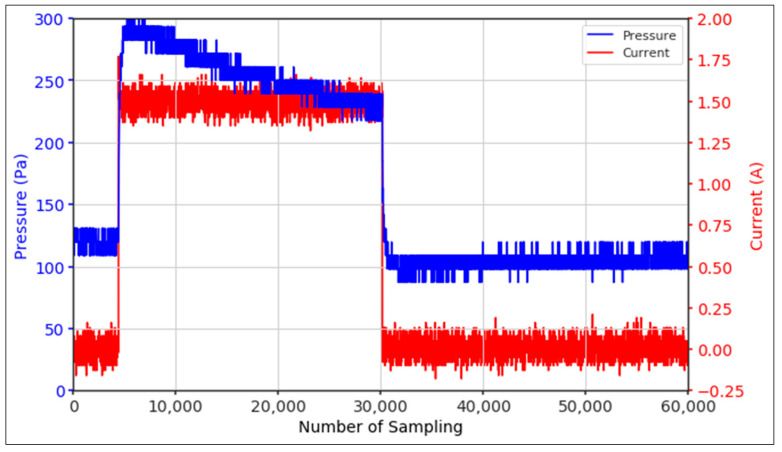
Pressure curve of the application of a magnetic field on for 30 s and off for 30 s.

**Table 1 micromachines-13-00222-t001:** Typical properties of MRF 122EG [34].

Appearance	Dark Gray Liquid
Viscosity, Pa-s @40 °C (104 °F)	0.042 ± 0.020
calculated as slope 500–800 s^−1^	
Density	
g/cm^3^	2.28–2.48
(lb/gal)	(19.0–20.7)
Solid content by weight, %	72
Flash point, °C (°F)	>150 (>302)
Operating temperature, °C (°F)	−40 to +130 (−40 to +266)

**Table 2 micromachines-13-00222-t002:** Comparison of the results of observations of pressure strength on variations in current and frequency of magnetic field application on the MRF.

Frequency	Current 0.5 A	Current 1.0 A	Current 1.5 A
On State	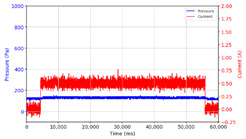	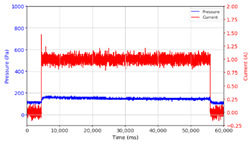	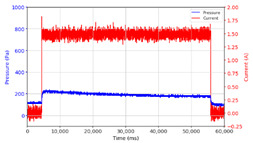
On–Off State	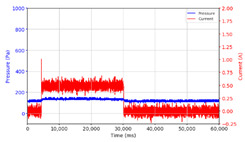	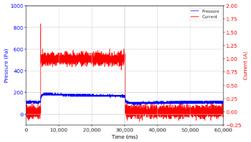	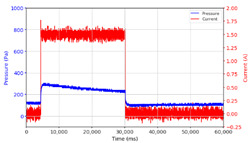
0.05 Hz	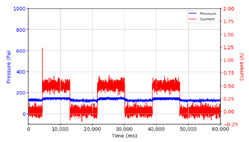	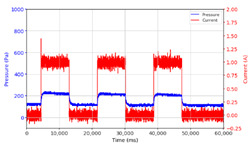	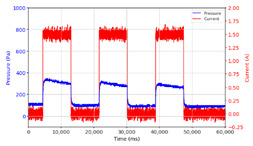
0.1 Hz	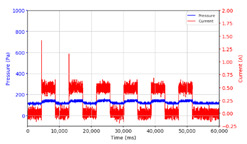	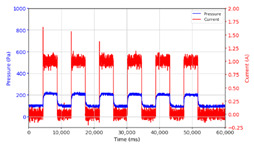	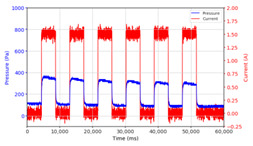
0.5 Hz	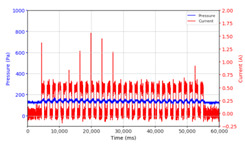	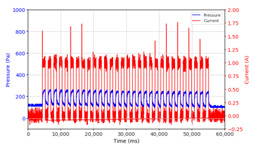	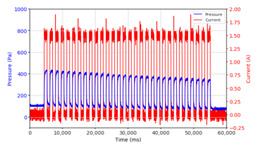
1 Hz	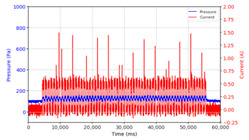	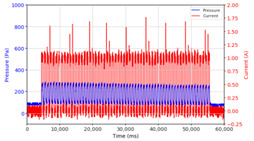	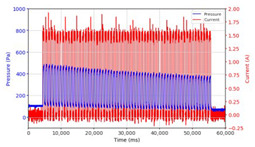
5 Hz	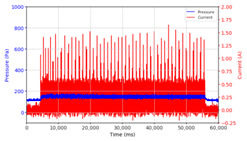	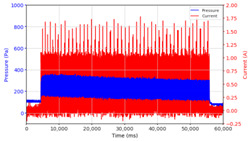	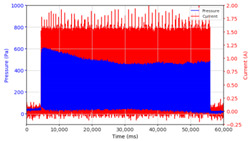
10 Hz	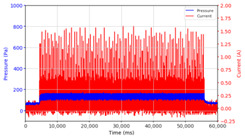	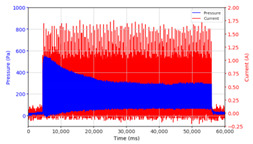	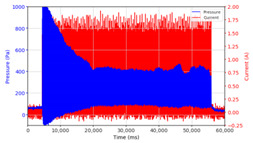

**Table 3 micromachines-13-00222-t003:** The results of continuous observations of the application of 1.5 A current at frequencies above 10 Hz on the MRF.

Frequency 20 Hz 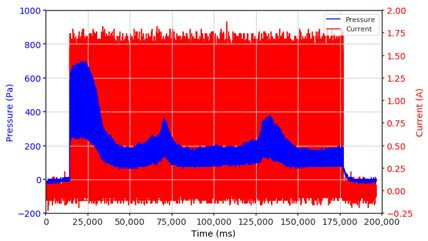	Frequency 40 Hz. 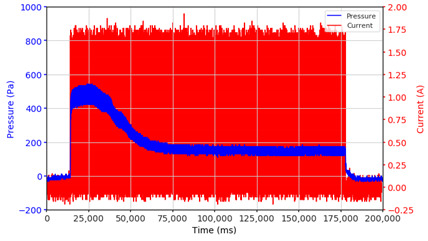
Frequency 50 Hz 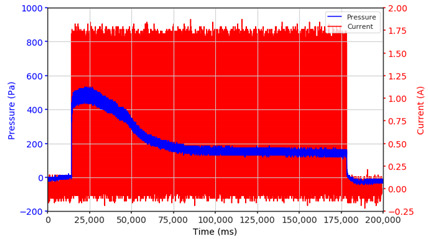	Frequency 100 Hz 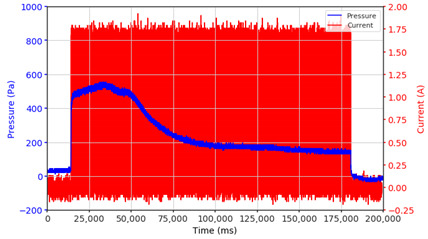

## Data Availability

Not applicable.

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
