# Peer review of "Effect of Time and Frequency of Magnetic Field Application on MRF Pressure Performance"

_micromachines, 2022, doi:10.3390/mi13020222_

Round 1

Reviewer 1 Report

Effect of time and frequency of magnetic field application on MRF pressure performance has been analyzed in this study. The work is interesting and meaningful. Detailed questions are as follows.

(1) More references about active actuator with MR fluid should be added in the section of Introduction.

(2) Fig. 3 shows the results of continuous observations for 60s. The responses in longer time is worth to be investigated.

(3) The effects of density of MR fluid and shape of particles on pressure performance should be investigated in the future.

Reviewer 2 Report

The authors have done an excellent investigation on the effect of Time and Frequency of Magnetic Field Application on Magnetorheological fluid Pressure Performance.

1 The AWG 24 coil thickness, turns details used in the simulation must be incorporated as a sentence in the manuscript

2. Throughout the article few grammar typo errors to be checked and corrected

3. Mention whether the number of turns of coil in both experiment and simulation is the same?
